# Direct gas-phase formation of formic acid through reaction of Criegee intermediates with formaldehyde

Pei-Ling Luo [1✉], I-Yun Chen[1], M. Anwar H. Khan [2] & Dudley E. Shallcross [2]

Ozonolysis of isoprene is considered to be an important source of formic acid (HCOOH), but its underlying reaction mechanisms related to HCOOH formation are poorly understood. Here, we report the kinetic and product studies of the reaction between the simplest Criegee intermediate ($CH_2OO$) and formaldehyde (HCHO), both of which are the primary products formed in ozonolysis of isoprene. By utilizing time-resolved infrared laser spectrometry with the multifunctional dual-comb spectrometers, the rate coefficient $k_{CH2OO+HCHO}$ is determined to be $(4.11 \pm 0.25) \times 10^{-12} \, cm^3 \, molecule^{-1} \, s^{-1}$ at 296 K and a negative temperature dependence of the rate coefficient is observed and described by an Arrhenius expression with an activation energy of $(-1.81 \pm 0.04) \, kcal \, mol^{-1}$. Moreover, the branching ratios of the reaction products $HCOOH + HCHO$ and $CO + H_2O + HCHO$ are explored. The yield of HCOOH is obtained to be 37–54% over the pressure (15–60 Torr) and temperature (283–313 K) ranges. The atmospheric implications of the reaction $CH_2OO + HCHO$ are also evaluated by incorporating these results into a global chemistry-transport model. In the upper troposphere, the percent loss of $CH_2OO$ by HCHO is found by up to 6% which can subsequently increase HCOOH mixing ratios by up to 2% during December-January-February months.

---

[1] Institute of Atomic and Molecular Sciences, Academia Sinica, Taipei 106319, Taiwan. [2] School of Chemistry, Cantock's Close, University of Bristol, Bristol BS8 1TS, UK. ✉email: plluo@gate.sinica.edu.tw

Being the simplest and most abundant organic acid in the atmosphere, formic acid (HCOOH) plays a crucial role in chemistry-climate interactions and to influence the atmospheric acidity. In the atmosphere, HCOOH can be produced from direct biogenic emissions, biomass burning as well as gas-phase and multi-phase chemical reactions; and it can be removed mainly through wet and dry deposition[1–6]. The mixing ratio of atmospheric HCOOH is typically observed from sub-100 pptv to a few ppbv levels and the lifetimes of HCOOH are estimated to be 1–2 days and 1–2 weeks in the boundary layer and upper troposphere, respectively[3,5]. Thanks to satellite observation techniques[1,5,6], the atmospheric abundance of HCOOH even in remote areas can be in-situ monitored. However, the current chemistry-climate models still cannot fully expound the unexpected high levels of observed HCOOH in the atmosphere. Typically, the HCOOH concentrations from modeling results are a factor of 2–5 times lower than that from observations[1,3,4,6]. Recently, the oxidation of gas-phase methanediol ($CH_2(OH)_2$), formed by volatilization of the hydrated formaldehyde in cloud droplets, has been proposed to be an important source of the atmospheric HCOOH[7,8], but it might not explain the high concentration of HCOOH observed in the cloud-free regions such as upper troposphere and lower stratosphere (UT/LS). On the other hand, the oxidation of isoprene has been widely considered as a major source of atmospheric formic acid[9,10]. According to the chamber experiments of isoprene oxidation[9], nearly 30% and up to 40% of the global annual production of atmospheric HCOOH from gas-phase reactions can be contributed from the OH-initiated oxidation of isoprene and isoprene ozonolysis, respectively. Exploring the explicit chemical mechanisms of the HCOOH formation from the isoprene oxidation is hence of interest in atmospheric chemistry for solving the issue of the underestimated abundance of HCOOH by the chemistry-climate models. In recent studies, the four-carbon Criegee intermediates such as methyl vinyl ketone oxide (MVKO) and methacrolein oxide (MACRO) have been reported to be the main products from the ozonolysis of isoprene and they can further undergo unimolecular dissociation to form OH and hydrocarbon radicals[11–13]. Besides the four-carbon Criegee intermediates, the simplest Criegee intermediate ($CH_2OO$) and formaldehyde (HCHO) are also the primary products generated, with yields of $(61 \pm 9)$ and $(81 \pm 16)\%$, respectively, in isoprene ozonolysis[13]. In the boundary layer and lower troposphere, the reaction of $CH_2OO$ with water dimers[14] has been recognized as the dominant sink for $CH_2OO$ and its primary products were recently determined to be hydroxymethyl hydroperoxide (HMHP) and HCHO with the branching ratios of $(55 \pm 15)$ and $(40 \pm 10)\%$, respectively[15]. Although the reaction of HMHP with OH has been proposed to produce HCOOH with a yield of $(45 \pm 14)\%$[16], it cannot explain the formation of HCOOH in the chamber experiments without OH scavengers[9]. Up to now, the reaction pathways related to the formation of HCOOH are barely understood, particularly for the environments with low concentrations of water and OH radicals.

According to the theoretical studies, HCOOH might be produced from the reaction of $CH_2OO$ with HCHO. Supplementary Fig. 1 shows the integrated enthalpy profiles of the reaction $CH_2OO + HCHO$ based on three theoretical investigations[17–19]. The detailed descriptions of the enthalpy profiles are presented in Supplementary Note 1. Jalan et al. employed the RCCSD(T)-F12a/VTZ-F12//B3LYP/ MG3S method and the Rice-Ramsperger-Kassel-Marcus (RRKM) theory to calculate the potential energy surface of the reaction $CH_2OO + HCHO$ and to investigate its rate coefficients and reaction products[17]. In their work, the HCOOH was proposed to be the major product of the reaction $CH_2OO + HCHO$ over a wide atmospheric

conditions and the stable secondary ozonide (SOZ) can only be formed at high pressure (>2 atm). The predicted rate coefficient was reported to be $8.3 \times 10^{-13}$ cm$^3$ molecule$^{-1}$ s$^{-1}$ at 298 K and a small negative temperature dependence was obtained and described by $k_{CH2OO+HCHO}$ $(T) = 4.5 \times 10^{-13}$ $\exp(0.36$ kcal mol$^{-1}/RT)$ cm$^3$ molecule$^{-1}$ s$^{-1}$ in a temperature range of $260 - 350$ K, where $R$ is the molar gas constant and $T$ is the temperature. In contrast to the predictions by Jalan et al.[17], Elakiya et al.[18] suggested that the formation of products $HCOOH + HCHO$ is unfavorable and the dominant product channel is the formation of $CO + H_2O + HCHO$. In a more recent theoretical study[19], Long et al. indicated that the rate coefficient of the reaction $CH_2OO + HCHO$ has a strong negative temperature dependence and the rate coefficients $k_{CH2OO+HCHO}$ $(T) = 2.7 \times 10^{-14} \exp(4.58$ kcal mol$^{-1}/RT)$ cm$^3$ molecule$^{-1}$ s$^{-1}$ were reported under high-pressure-limit and at $280 - 350$ K. However, the theoretical results of the $k_{CH2OO+HCHO}$ from Jalan et al.[17] and Long et al.[19] show a significant discrepancy. The 298 K rate coefficient reported by Long et al., $6.2 \times 10^{-11}$ cm$^3$ molecule$^{-1}$ s$^{-1}$, was ~75 times larger than the value predicted by Jalan et al., $8.3 \times 10^{-13}$ cm$^3$ molecule$^{-1}$ s$^{-1}$. Up to now, no experimental studies have been conducted to evaluate the kinetics and product yields of this potentially interesting reaction.

Herein, we report direct measurements of the rate coefficients and products of the reaction between $CH_2OO$ and HCHO by employing mid-infrared laser systems coupled with a Herriott-type flash photolysis cell. The experimental approaches are described in Methods and illustrated in Supplementary Fig. 2. In addition to investigation of the temperature and pressure dependencies of the rate coefficients, we determine the branching ratios of primary product channels, $HCOOH + HCHO$ and $CO + H_2O + HCHO$ in the reaction $CH_2OO + HCHO$ via simultaneous determination of HCOOH and CO using synchronized two-color time-resolved dual-comb spectroscopy at varied experimental conditions. The atmospheric implications of the reaction $CH_2OO + HCHO$ are also evaluated by employing a global 3-D chemistry-transport model, STOCHEM-CRI. Our results may explain the observations of HCOOH in the simulation chamber and remote atmosphere at UT/LS height levels.

## Results

**Kinetic measurements of the reaction $CH_2OO + HCHO$.** To determine the rate coefficients of the reaction between $CH_2OO$ and HCHO and to evaluate their temperature and pressure dependencies, we first performed the kinetic measurements under different total pressures at 296 K. Over 63 measurements in seven sets were carried out at experimental conditions with $[CH_2OO]_0 = (5.5–8.6) \times 10^{12}$ molecules cm$^{-3}$, $[HCHO]_0 = (1.8–34.5) \times 10^{14}$ molecules cm$^{-3}$, and total pressure $P_T$ = 6.4–56.0 Torr at 296 K. For kinetic studies of the reaction $CH_2OO + HCHO$ under pseudo-first-order conditions ($[HCHO] \gg [CH_2OO]$), the decay in the concentration of $CH_2OO$ with time can be described using a single-exponential formula:

$$[CH_2OO]_{obs}(t) = [CH_2OO]_0 \times \exp(-k_{obs} \times t) \quad (1)$$

in which $[CH_2OO]_{obs}$ (t) represents the observed $CH_2OO$ concentration as a function of time, $[CH_2OO]_0$ represents the initial concentration of $CH_2OO$, and $k_{obs}$ represents the overall decay rate coefficient for the observed $CH_2OO$:

$$k_{obs} = k_0 + k_{CH2OO+HCHO} \times [HCHO]_0 \quad (2)$$

where $k_0$ is the decay rate coefficient contributed by the $CH_2OO$ self-reaction and reactions of $CH_2OO$ with other species generated in the reaction system, $k_{CH2OO+HCHO}$ is the second-order rate coefficient of the reaction $CH_2OO + HCHO$, and $[HCHO]_0$ is the initial concentration of formaldehyde.

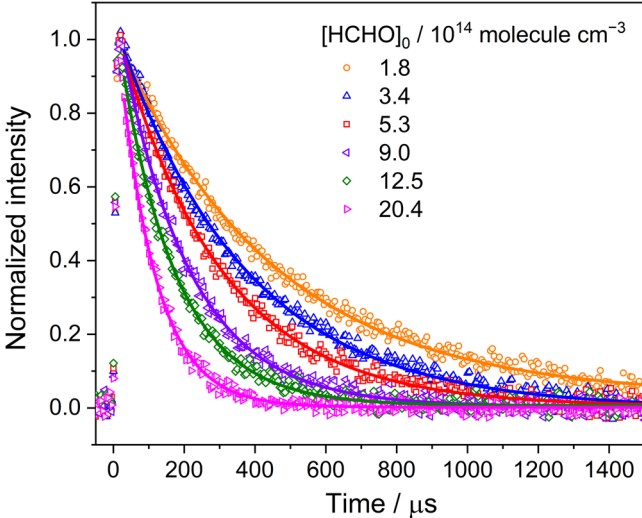

**Fig. 1 Temporal profiles of the $CH_2OO$ obtained under conditions with different $[HCHO]_0$.** The open symbols represent time-resolved signals of a $CH_2OO$ absorption line at 1285.611 cm$^{-1}$ recorded with a time resolution of 5 μs. The solid lines represent fitting curves. Each temporal profile was fitted with a single-exponential function to obtain the decay rate coefficient ($k_{obs}$). The data correspond to experiment 1–6 listed in Supplementary Table 1 with total pressure $P_T = 13.3$ Torr, temperature 296 K, and $[CH_2OO]_0 \approx 7.4 \times 10^{12}$ molecule cm$^{-3}$. The derived $k_{obs}$ as a function of $[HCHO]_0$ are shown in Supplementary Fig. 4.

Figure 1 shows the representative temporal profiles of the $CH_2OO$ measured at 1285.611 cm$^{-1}$ under varied initial concentrations of HCHO, $(1.8–20.4) \times 10^{14}$ molecules cm$^{-3}$, at a total pressure of 13.3 Torr and 296 K. Each observed profile (open symbol) was fitted with single-exponential curve (solid line) to derive the decay rate coefficient ($k_{obs}$). In addition, we also analyzed the time traces of $CH_2OO$ with a kinetic model, as shown in Supplementary Fig. 3. The kinetic model, taking into account key reaction paths including the formation and self-reaction of $CH_2OO$ as well as the $CH_2OO + I$ reaction[20], have been used in other reaction kinetic studies such as $CH_2OO + NO_2$[21] and $CH_2OO + HCl$[22]. The obtained first-order rate coefficients from the single-exponential and model fits as a function of $[HCHO]_0$ are displayed in Supplementary Fig. 4. The second-order rate coefficient $k_{CH2OO+HCHO}$, corresponding to the fitted slopes derived by using single-exponential and model fits, were consistent with each other, supporting the feasibility of kinetic analysis under pseudo-first-order conditions.

Figure 2 depicts the obtained $k_{CH2OO+HCHO}$ as a function of the total pressure at 296 K and the rate coefficients were observed to be pressure independent under the presented experimental conditions. A summary of experimental conditions and obtained $k_{obs}$ rate coefficients of all kinetic measurements are listed in Supplementary Table 1. Considering one standard deviation of the $k_{CH2OO+HCHO}$ obtained from 7 experimental sets (2%) and the errors in determination of $[HCHO]_0$ (4%), flow rates (3%), temperature (1%), and pressure (3%), the overall standard error was estimated to be ~6% and the $k_{CH2OO+HCHO}$ was hence obtained to be $(4.11 \pm 0.25) \times 10^{-12}$ cm$^3$ molecule$^{-1}$ s$^{-1}$ at 296 K. Moreover, the temperature dependence of the rate coefficient was also investigated over the temperature range 268.6–336.5 K. Figure 3 shows the representative plots of $k_{obs}$ vs. $[HCHO]_0$ at six different temperatures. The slopes of the plots of $k_{obs}$ vs. $[HCHO]_0$ were found decreasing with temperature. The temperature dependence of the rate coefficients $k_{CH2OO+HCHO}$ is

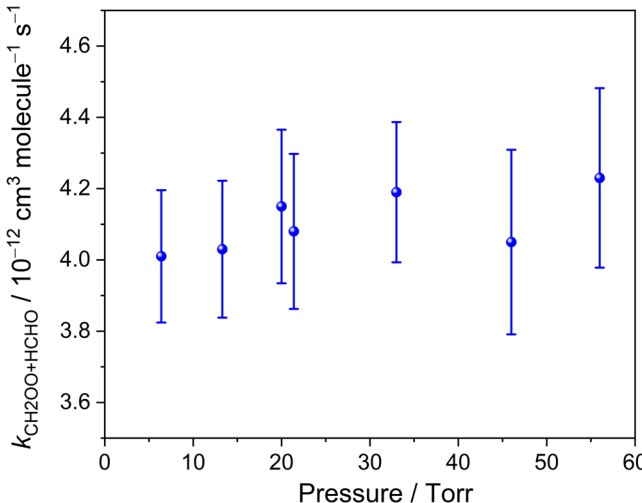

**Fig. 2 Rate coefficients for the reaction $CH_2OO + HCHO$ as a function of the total pressure.** Each rate coefficient is the fitted slope of the plot of $k_{obs}$ against $[HCHO]_0$ of each experimental set at 296 K. The error bars include the errors of the fitted slope and determined $[HCHO]_0$.

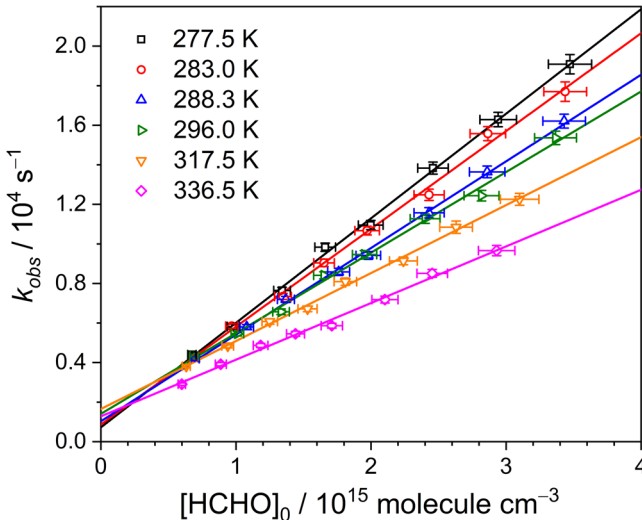

**Fig. 3 Plots of $k_{obs}$ vs. $[HCHO]_0$ at six different temperatures.** The open symbols represent experimental data and the solid lines represent linear fitting curves. The data correspond to experiment sets 7, 8, 10, 12, 14, and 15 listed in Supplementary Table 1 with total pressure $P_T = 21.3–21.4$ Torr. The error bars represent the error of each $k_{obs}$ obtained by fitting each $CH_2OO$ temporal absorbance profile and the uncertainty of the determined $[HCHO]_0$.

shown in Fig. 4 and it was fitted with the Arrhenius expression:

$$k_{CH2OO+HCHO}(T) = A \times \exp(-E_a/RT) \qquad (3)$$

where A is the pre-exponential constant, R is the molar gas constant, and $E_a$ represents the activation energy for the reaction $CH_2OO + HCHO$. The A and $E_a$ were, respectively, obtained to be $(1.91 \pm 0.15) \times 10^{-13}$ cm$^3$ molecule$^{-1}$ s$^{-1}$ and $(-1.81 \pm 0.04)$ kcal mol$^{-1}$, with an 1σ statistical error of fitting. A comparison of experimental and theoretical results of the rate coefficient for the reactions of $CH_2OO$ with HCHO, $CH_3CHO$, and $CH_3COCH_3$ is shown in Supplementary Table 2. Both experimental and theoretical results indicate that rate coefficients decreases in the order $k_{CH2OO+HCHO} > k_{CH2OO+CH3CHO} > k_{CH2OO+CH3COCH3}$[17,23,24]. The obtained $k_{CH2OO+HCHO}$ in this work is 15 times smaller than the theoretical value reported by

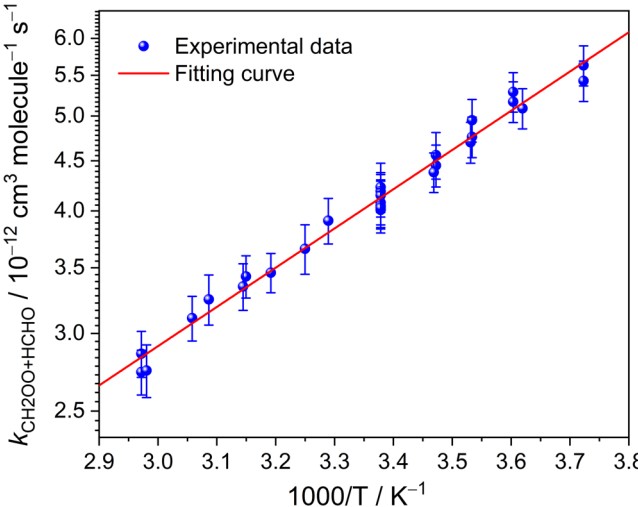

**Fig. 4 Arrhenius plot of the rate coefficients for the reaction $CH_2OO + HCHO$.** The blue circles represent experimental data and all of data are fitted (red solid line) with the Arrhenius expression, $k_{CH2OO+HCHO}$ $(T) = (1.91 \pm 0.15) \times 10^{-13} \exp[(1.81 \pm 0.04)$ kcal mol$^{-1}$/RT]. Here, each rate coefficient is the fitted slope of the plot of $k_{obs}$ against [HCHO]$_0$ of each experimental set at different conditions. The error bars include the errors of the fitted slope and determined [HCHO]$_0$.

Long et al.[19], but it is ~5 times larger than the predicted value from Jalan et al.[17] Although this activation energy for $CH_2OO +$ HCHO is not in agreement with the theoretical results, it is comparable with $(-2.2 \pm 0.7)$ kcal mol$^{-1}$ measured by Elsamra et al.[23] for reactions involving other ketones: $CH_2OO + CH_3CHO$ and $CH_2OO + CH_3COCH_3$.

**Branching yields of HCOOH and CO.** According to the theoretically computed enthalpy profiles, as shown in Supplementary Fig. 1, the reaction of $CH_2OO$ with HCHO is expected to lead to the formation of HCOOH + HCHO or $CO + H_2O +$ HCHO. To evaluate the branching yields of the two potential product channels, we empolyed two sets of dual-comb spectrometers to simultaneously measure the time-resolved absorption spectra of the HCOOH and CO near 8.9 and 4.5 μm, respectively. Figure 5 shows the representative time-resolved dual-comb spectra in the regions 1127.93−1128.54 and 2218.64−2219.09 cm$^{-1}$ with the spectral sampling spacing of 288 MHz ($9.6 \times 10^{-3}$ cm$^{-1}$) and the temporal resolution of 12 μs. With the addition of HCHO in the reaction system of $CH_2I + O_2$, a significant formation of HCOOH was observed, as shown in Fig. 5a. By contrast, no obvious absorption signal of HCOOH could be found for the experiment in the absence of HCHO, as shown in Fig. 5c. In addition, the strong absorption signals of a CO transition line at 2218.7455 cm$^{-1}$ were observed in both experiments with and without the addition of HCHO, as shown in Fig. 5b, d, respectively. Several weak absorption peaks were also found in Fig. 5b, d at very early reaction time and they could be tentatively assigned to the absorption of HCO radical. The HCO radical might be generated from the decomposition of the simplest Criegee intermediates and it could be further reacted away by excess $O_2$ to form $HO_2 + CO^{25,26}$.

By employing the high-resolution time-resolved dual-comb spectroscopy, the time-dependent concentrations of HCOOH and CO can be estimated by analyzing their rotationally resolved infrared absorption spectra. To estimate the concentration of the HCOOH in the reaction system, several transition lines of the $\nu_6$ band of the *trans*-HCOOH near 1128 cm$^{-1}$ were measured with dual-comb spectroscopy and analyzed with well-characterized

spectral parameters[27–29], as shown in Supplementary Fig. 5. Considering the uncertainty of line parameters of HCOOH (7%) and the standard deviation of residuals between observed and simulated spectra (7%), the uncertainty for each time-dependent concentration of HCOOH was estimated to be 10%. In each experiment, the CO was monitored using the high-resolution dual-comb spectra near 4.5 μm, as displayed in Supplementary Fig. 6. The concentration of CO was quantified by measuring a fundamental transition line, R(21) at 2218.7455 cm$^{-1}$, with the line strength of $2.855 \times 10^{-20}$ cm molecule$^{-1}$ [29]. By taking account into the uncertainty of the CO line strength (1%) and the analyzed error of the difference absorbance spectra (7%), the uncertainty of ~7% can be obtained for each time-dependent CO concentration.

To further determine the product yields of the reaction $CH_2OO + HCHO$, a simple kinetic model, as shown in Supplementary Table 3, was adopted to simulate the temporal concentration profiles of HCOOH and CO that formed from the reaction $CH_2OO + HCHO$. Based on our experiments and the theoretical predictions, three product channels were considered in the model:

$$CH_2OO + HCHO \rightarrow HCOOH \,(\text{cold and energized})$$
$$+ \, HCHO \,(R_{1a} \text{ and } R_{1a'})$$

$$CH_2OO + HCHO \rightarrow CO(\nu \geq 0) + H_2O + HCHO \,(R_{1b} \text{ and } R_{1b'})$$

$$CH_2OO + HCHO \rightarrow \text{other products} \,(R_{1c})$$

in which the branching ratios of the HCOOH (cold and energized) + HCHO and CO $(\nu \geq 0) + H_2O +$ HCHO product channels are $y_{HCOOH}$ and $y_{CO}$, respectively, and the branching ratio of other products is $1- y_{HCOOH} - y_{CO}$. In addition, the relative yields of the initially cold HCOOH and energized HCOOH$^\#$ were set to be α and $1- α$, where the cold HCOOH corresponds the *trans*-HCOOH ($\nu = 0$) and HCOOH$^\#$ may include the *trans*-HCOOH ($\nu > 0$) and the higher-energy *cis* conformer of formic acid. The relaxations of the energized HCOOH$^\#$, including *trans*-HCOOH ($\nu > 0$) to *trans*-HCOOH ($\nu = 0$) and the *cis*-HCOOH to *trans*-HCOOH conversions[30] were also taken into account in the model. Figure 6 shows the comparison of the measured and simulated temporal profiles of HCOOH. A fast formation rate of the initially cold HCOOH was observed almost corresponding to the decay rate of $CH_2OO$ and the slow formation of the part of HCOOH was observed with the rates of 100−200 s$^{-1}$. On the other hand, the reaction of $CH_2OO$ with HCOOH was also considered in the model because of the fast rate coefficients of the reaction $CH_2OO +$ HCOOH. The temperature-dependent rate coefficients of the reaction $CH_2OO +$ HCOOH have been determined by Peltola et al.[31] and the rate coefficient is $(1.0 \pm 0.03) \times 10^{-10}$ cm$^3$ molecule$^{-1}$ s$^{-1}$ at 296 K. We assumed that the rate coefficients of the reactions $CH_2OO +$ HCOOH and $CH_2OO +$ HCOOH$^\#$ are the same. By adding these reactions in the model, we observed that the $y_{HCOOH}$ increased by ~10% for experimental conditions with $[CH_2OO]_0 = (3.2–3.8) \times 10^{13}$ molecules cm$^{-3}$, $[HCHO]_0 = (2.2–4.0) \times 10^{15}$ molecule cm$^{-3}$, and $T = 283−313$ K. Figure 7 shows the comparison of the measured and simulated temporal profiles of CO. Because the CO can be formed from the direct decomposition of the $CH_2OO$ and the reaction of $HCO + O_2^{25,26}$, we carried out the measurements by replacing HCHO to $SO_2$ to obtain the concentration profile of CO generated from instant decomposition of the energized $CH_2OO^*$. Additionally, the CO formed from the photodissociation of HCHO by 248-nm photolysis beam was also taken into account in each experiment set. The fractional yield of photodissociation of HCHO was obtained to be ~0.02% at 248 nm and approximately

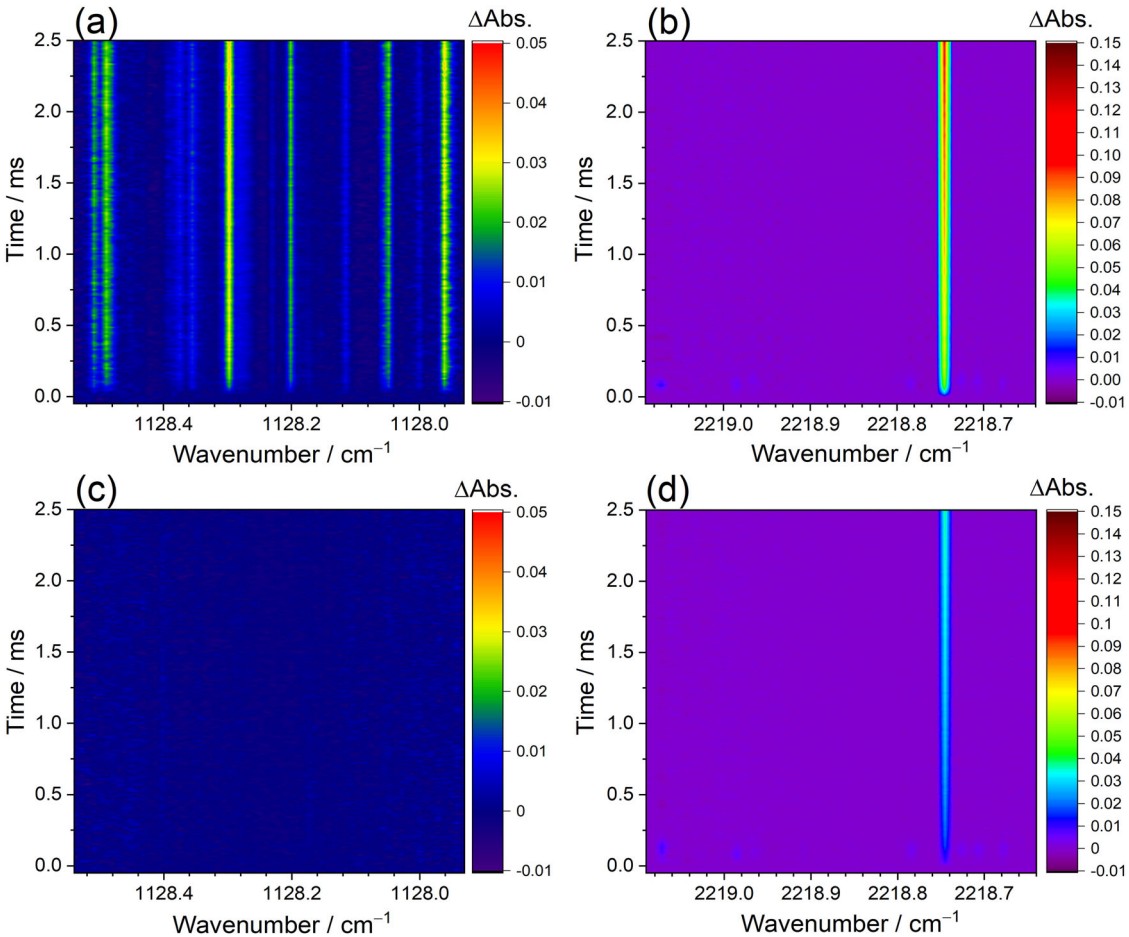

**Fig. 5 Time-resolved dual-comb spectra of HCOOH and CO.** The spectra were measured in the regions 1127.93−1128.54 cm$^{-1}$ (**a**) and (**c**) and 2218.64−2219.09 cm$^{-1}$ (**b**) and (**d**). The spectra **a** and **b** were recorded simultaneously after the 248-nm irradiation of a flowing mixture of CH$_2$I$_2$/HCHO/ O$_2$/N$_2$ (0.016/0.123/14.9/0.04 Torr, $P_T$ = 15.1 Torr, 296 K) over 5000 excimer laser shots. The spectra **c** and **d** were measured simultaneously after 248-nm laser photolysis of the flowing mixture of CH$_2$I$_2$/O$_2$/N$_2$ (0.016/14.9/0.04 Torr, $P_T$ = 15.0 Torr, 296 K) over 5000 excimer laser shots. Here, the spectral sampling spacing is 288 MHz (9.6 × 10$^{-3}$ cm$^{-1}$) and the temporal resolution is 12 µs.

$8 \times 10^{11}$ molecule cm$^{-3}$ of CO could be produced when the [HCHO]$_0$ of $4 \times 10^{15}$ molecule cm$^{-3}$ was used. Figure 7e indicates the corrected concentration profile of CO that represents the net production of CO by the reaction CH$_2$OO + HCHO. To analyze the concentration profile of CO with the kinetic model, the relative yields of the cold CO and the vibrationally excited CO$^\#$ ($v > 0$) were set to be β and 1− β. Furthermore, the relaxation of CO$^{\#}$ [32] were also taken into account in the mode. A summary of experimental conditions, fitting parameters, and the derived branching ratios (y$_{HCOOH}$ and y$_{CO}$) is listed in Supplementary Table 4. Considering the uncertainty of [CH$_2$OO]$_0$ (10%), analyzed errors of the temporal concentration profiles (5%), and the uncertainties of each time-dependent concentration for HCOOH (10%) and CO (7%), we estimated the overall uncertainty of y$_{HCOOH}$ and y$_{CO}$ to be 15% and 13%, respectively. Figure 8 illustrates the pressure dependence of branching ratios for the primary product channels of the reaction CH$_2$OO + HCHO at 283 K, 296 K, and 313 K. The yields of HCOOH were found slightly increasing with pressure and decreasing with temperature. In comparison, the CO yields show a negative pressure dependence and the pressure-dependent variations become smaller at higher temperature. At the total pressure of 15.1 Torr at 296 K, the branching ratios for the formation of products HCOOH + HCHO (y$_{HCOOH}$) and CO + H$_2$O + HCHO (y$_{CO}$) were determined to be 0.43±0.06 and 0.57±0.07, respectively, indicating that all of CH$_2$OO were converted into HCOOH and CO + H$_2$O by HCHO. Most

significantly, the HCOOH can be directly generated from the reaction CH$_2$OO + HCHO with the high yields of 37–54 % over the pressure range 15–60 Torr and temperature range 283–313 K. This work indicates that the reaction CH$_2$OO + HCHO might make a significant contribution to the HCOOH formation under wide atmospheric conditions.

**Discussion**

The atmospheric importance of the reaction of CH$_2$OO with HCHO depends upon its relatively scavenging ability to compete with the consumptions of CH$_2$OO by water vapor[33] as well as the HCHO by the OH radicals[34], affected by relative humidity (RH), temperature and photochemical conditions. Supplementary Fig. 7 shows the fractional contribution of the reactions, CH$_2$OO + HCHO, CH$_2$OO + H$_2$O, and CH$_2$OO + ( H$_2$O)$_2$, to the total loss rate of CH$_2$OO over a relative humidity (RH) range of 1–50% at temperatures of 296, 273, and 250 K while the mixing ratios of HCHO are set to be 0.1 ppmv (Supplementary Fig. 7a) and 1 ppmv (Supplementary Fig. 7b) at 1 atm. In the ambient atmosphere, the mixing ratios of gaseous HCHO can be observed up to sub-ppmv and a few ppmv levels, respectively, in outdoor and indoor environments[35,36]. With the reported mean concentration of HCHO in schools in Anhui (~1.4 ppm)[36] and rate coefficient of CH$_2$OO + HCHO at 296 K, the effective first-order rate coefficient is estimated to be

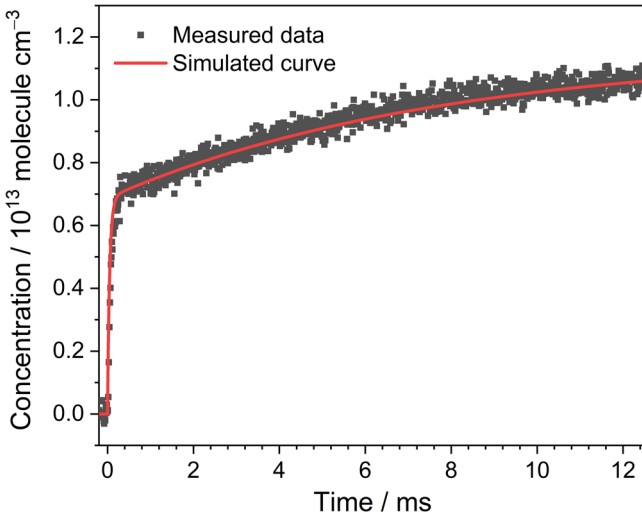

**Fig. 6 Temporal concentration profiles of HCOOH.** The black square represents measured temporal profile with a time resolution of 12 μs. The red solid line represents simulation profile using the kinetic model shown in Supplementary Table 3. Here, $[CH_2OO]_0 = 3.71 \times 10^{13}$ molecule $cm^{-3}$, $[HCHO]_0 = 4.0 \times 10^{15}$ molecule $cm^{-3}$, $P_T = 15.1$ Torr, and T = 296 K. Here, the data correspond to the experiment 1 listed in Supplementary Table 4.

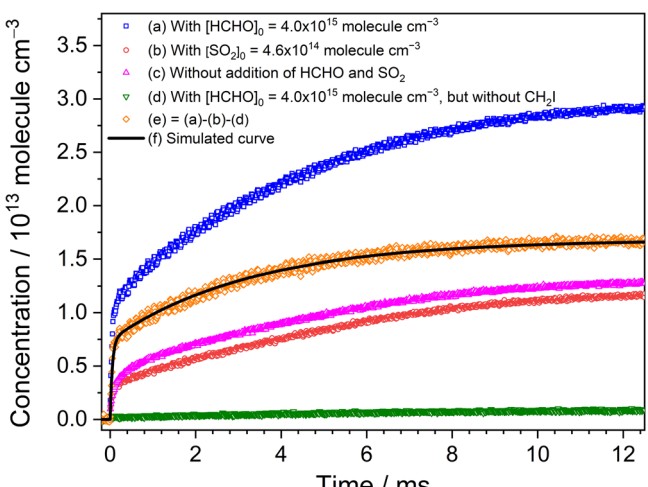

**Fig. 7 Temporal concentration profiles of CO. a** The concentration profile of CO was measured after 248-nm laser photolysis of the flowing mixture of $CH_2I_2/HCHO/O_2/N_2$ (0.016/0.123/14.9/0.04 Torr, $P_T = 15.1$ Torr, 296 K). **b** The concentration profile of CO was measured after 248-nm laser photolysis of the flowing mixture of $CH_2I_2/SO_2/O_2/N_2$ (0.016/0.014/14.9/0.04 Torr, $P_T = 15.0$ Torr, 296 K). **c** The concentration profile of CO was measured after 248-nm laser photolysis of the flowing mixture of $CH_2I_2/O_2/N_2$ (0.016/14.9/0.04 Torr, $P_T = 15.0$ Torr, 296 K). **d** The concentration profile of CO was measured after 248-nm laser photolysis of the flowing mixture of $HCHO/O_2/N_2$ (0.123/14.9/0.04 Torr, $P_T = 15.0$ Torr, 296 K). **e** The corrected concentration profile of CO represents the net production of CO by the reaction $CH_2OO + HCHO$. **f** The concentration profile simulated using the kinetic model shown in Supplementary Table 3. Here, the data correspond to the experiment 1 listed in Supplementary Table 4.

$\sim143\,s^{-1}$ which is comparable to that $(\sim123\,s^{-1})$ of $CH_2OO$ with water monomer at RH of 50% and 296 K, indicating the potential importance of the reaction $CH_2OO + HCHO$ in indoor chemistry. At low temperature of 250 K (−23 °C), we found that the reaction $CH_2OO + HCHO$ might compete well

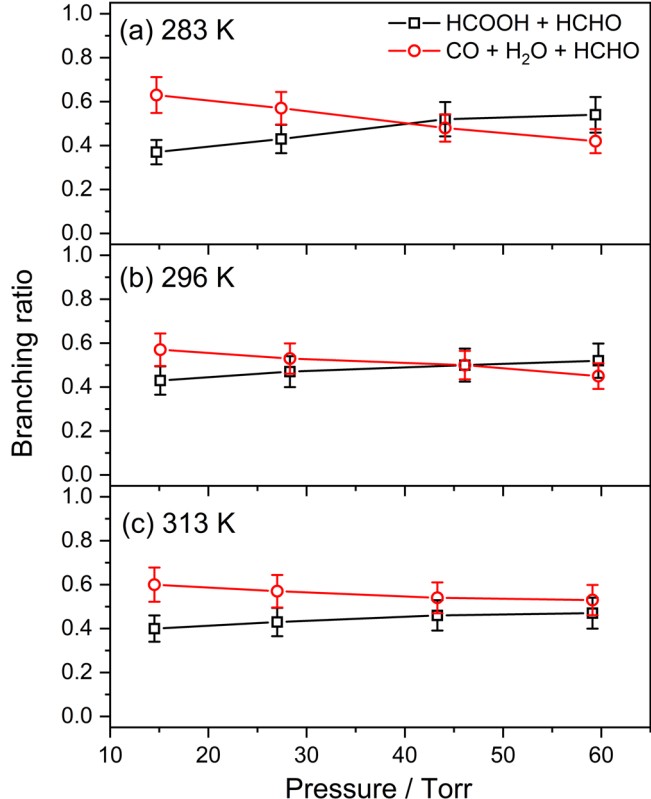

**Fig. 8 Product branching ratios at various pressures and temperatures.** The pressure dependences of branching ratios for products of the reaction $CH_2OO + HCHO$ were obtained at **a** 283 K, **b** 296 K, and **c** 313 K. The error bars represent the overall uncertainties of $y_{HCOOH}$ (black) and $y_{CO}$ (red), that include the uncertainty of determined $[CH_2OO]_0$, analyzed errors of the temporal concentration profiles, and the uncertainties of each time-dependent concentration for HCOOH and CO, respectively.

with the reaction $CH_2OO + (H_2O)_2$ for the removal of $CH_2OO$ over the RH range below 25%, suggesting that the reaction $CH_2OO + HCHO$ might also play a crucial role in the high-altitude atmosphere such as upper troposphere and lower stratosphere (UT/LS).

The model simulation results show that the reaction, $HCHO + CH_2OO$ has little impact on modeled HCOOH surface level with increasing its level by up to 0.1% at the forested regions (Fig. 9). Compared with surface level, the concentration of $CH_2OO$ was found to be higher (~400 molecules $cm^{-3}$) at upper troposphere because of much lower water concentartions[37]. Thus the $CH_2OO + HCHO$ reaction can compete with the reaction $CH_2OO + (H_2O)_2$ at the upper troposphere. Figure 10 shows the annual zonal percentage loss of $CH_2OO$ by HCHO and the annual zonal percentage changes in HCOOH mixing ratios. The model results show that the loss of $CH_2OO$ by HCHO can contribute by up to 1% and 6% to the total loss of $CH_2OO$ at the upper troposphere during June–July–August and December–January–February, respectively, as shown in Fig. 10a, b. The nonnegligible loss of $CH_2OO$ by HCHO at the upper troposphere resulted in an increment of HCOOH mixing ratios by up to 2% during December–January–February months (Fig. 10d). The results can explain the high gas-phase mixing ratios of HCOOH observed in the high Arctic[2] and from the satellite observations at the altitude range of 10–16 km[38], as shown in Supplementary Fig. 8.

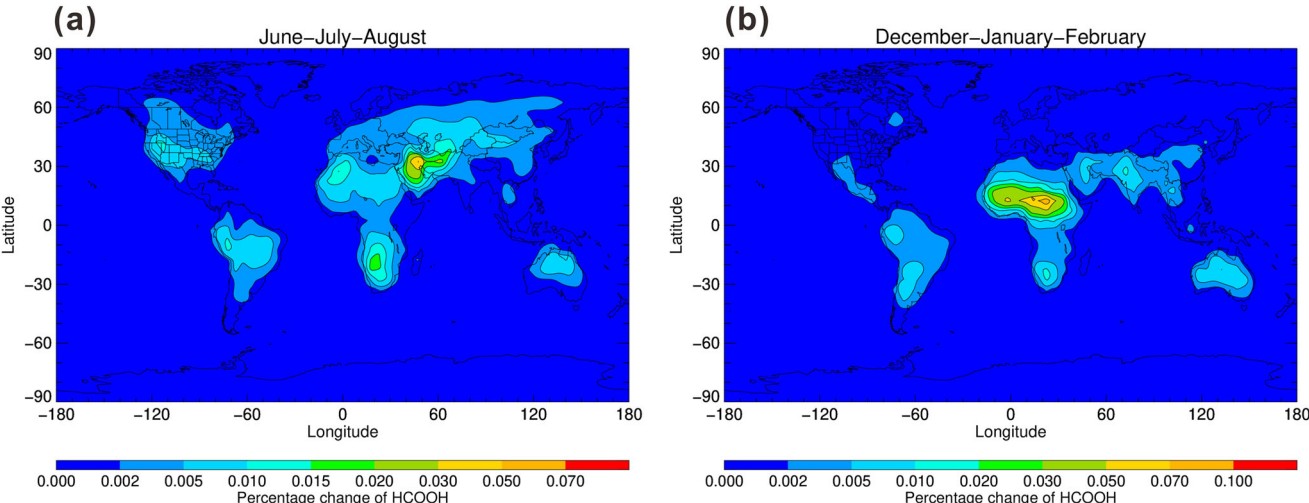

**Fig. 9 Annual surface percentage changes in HCOOH mixing ratios. a** During June–July–August and **b** during December–January–February, on inclusion of the HCHO + CH₂OO reaction compared with the base case model.

**Fig. 10 Annual zonal percentage loss of CH₂OO by HCHO and percentage changes in HCOOH mixing ratios.** The annual zonal percentage loss of CH₂OO by HCHO **a** during June–July–August and **b** during December–January–February; the annual zonal percentage changes in HCOOH mixing ratios **c** during June–July-August and **d** during December–January–February, on inclusion of the HCHO + CH₂OO reaction compared with the base case model.

In the model, CH₂OO is produced only from ethene, propene, isoprene and β-pinene (assuming 33% of the total monoterpenes). But many of the monoterpenes and sesquiterpenes in the MEGAN model are CH₂OO precursors on ozonolysis[39]. Including the unaccounted flux of reactive biogenic and anthropogenic terminal alkenes in STOCHEM-CRI model would result in significantly higher concentrations of CH₂OO than predicted by current model setting, and would

increase the modeled impact of the loss of $CH_2OO$ by HCHO herein.

## Conclusion

In conclusion, we have performed kinetic and product determinations of the reaction of the simplest Criegee intermediate $CH_2OO$ with HCHO by employing the highly flexible mid-infrared comb lasers and synchronized two-color time-resolved dual-comb spectroscopy. The bimolecular rate coefficients for the reaction $CH_2OO$ + HCHO were evaluated over the temperature range 268.6–336.5 K at total pressure of 6.4–56.0 Torr.

The rate coefficient of $(4.11 \pm 0.25) \times 10^{-12}$ $cm^3$ molecule$^{-1}$ s$^{-1}$ at 296 K was determined and the temperature-dependent rate coefficients were obtained and described using an Arrhenius expression with an activation energy of $(-1.81 \pm 0.04)$ kcal mol$^{-1}$. Furthermore, the branching yields of the products HCOOH + HCHO and CO + $H_2O$ + HCHO in the reaction $CH_2OO$ + HCHO were evaluated by direct quantifications of HCOOH and CO with rotationally resolved infrared absorption spectroscopy. The branching ratios of the HCOOH and CO product channels were determined to be 37–54 % and 42–63 % under the experimental conditions with pressure of 15–60 Torr and temperature of 283–313 K, indicating that $CH_2OO$ could be fully converted to HCOOH and CO + $H_2O$ in the presence of excessive HCHO. These results could explain the high HCOOH levels observed in the chamber experiments of isoprene oxidation under low relative humidity (RH < 5%) and in the absence of the OH scavengers[9]. The 3-D global chemistry-transport modeling results show that the loss of $CH_2OO$ by HCHO is nonnegligible which can increase HCOOH level by 2% in the upper troposphere during December–January–February months. Overall, the reaction $CH_2OO$ + HCHO might have potential implications in the atmospheric chemistry of both outdoor and indoor environments and would make an important contribution to the HCOOH formation in atmosphere at the upper troposphere. Updating of model alkene inventory is anticipated to demonstrate that $CH_2OO$ + HCHO reaction can have an even greater impact on HCOOH formation than our current suggested assessment.

## Methods

**Highly flexible mid-infrared comb lasers and synchronized two-color time-resolved dual-comb spectroscopy.** Schematic of the experimental approaches for the kinetic and product studies of the reaction of the simplest Criegee intermediate ($CH_2OO$) with formaldehyde (HCHO) is illustrated in Supplementary Fig. 2. The mid-infrared comb lasers with switchable dual-comb and continuous-wave (cw) operation modes, as the probing beams were coupled into a Herriott reactor cell. The mid-infrared comb lasers were established based on difference frequency generation of the near infrared electro-optic comb lasers and the tunable cw lasers[40,41], thus the operating spectral region can be flexibly adjusted and expanded. In the kinetic measurements, we employed the mid-infrared laser with the cw operation mode at 7.8 μm to probe the $CH_2OO$ sensitively. After passing through the Herriot cell, the mid-infrared laser was detected using a liquid-nitrogen-cooled HgCdTe detector (MCT) and the time-resolved signals were recorded by a data-acquisition board (DAQ). The minimal detectable concentration of $CH_2OO$, $[CH_2OO] \approx 8 \times 10^{10}$ molecule cm$^{-3}$, was estimated by recording the temporal profiles of the $CH_2OO$ absorption line at 1285.611 cm$^{-1}$ with a time resolution of 5 μs and 200 averages. This level of sensitivity can allow us to perform the kinetic measurements using low concentration of $CH_2OO$ and to implement kinetic studies of the reaction $CH_2OO$ + HCHO under pseudo-first-order conditions ([HCHO]»[$CH_2OO$]). Furthermore, another two sets of dual-comb lasers with central wavelengths, respectively, near 8.9 and 4.5 μm were used to determine the formic acid (HCOOH) and carbon monoxide (CO), both of which are theoretically predicted products of the reaction between $CH_2OO$ and HCHO. To record time-resolved dual-comb spectroscopy before and after laser photolysis of precursor mixtures, both the excimer laser and DAQ were synchronized with the multi-heterodyne signals of the dual-comb laser. With high-resolution time-resolved dual-comb spectroscopy, the spectral sampling spacing is corresponding to the repetition frequency of the employed comb laser and the spectral sampling points can be increased by interleaving dual-comb spectra measured with different spectral sampling spacings[26,42]. In addition, the concentration of each probed species can be estimated by the obtained absorbance spectra and the line strengths

of the molecular transitions taken from the database[29]. Moreover, the temporal resolution in the time-resolved measurements can be simply adjusted from μs to ms level by setting the length of dual-comb interferogram used for generation of each time-dependent Fourier transform (FT) spectrum. Therefore, we can analyze the same experimental raw data with different time-resolution to obtain the best quality of the time-resolved spectra.

**Herriott reactor cell and precursor preparation.** To monitor the transient molecules generated upon laser photolysis of precursor mixtures, the Herriott-type reactor cell was constructed based on a pair of 2-inch concave gold mirrors and a double-layer glass tube. The mirrors were designed with a radius of curvature of 400 mm, a 25-mm-diameter center hole allowed passage of the excimer laser as the photolysis beam, and a 4-mm-diameter off-axis hole served as the entrance and exit of the mid-infrared probe beams. With the Herriott multipass cell, an optical path length of the probe beam was determined to be 41.8 m and the overlapped path between the photolysis and probe beams was estimated to be 13.4 m. The double-layer glass tube as the main body of the reactor can be temperature controlled from 268 to 338 K by a liquid circulator. The main stream of the flowing mixtures was also pre-cooled or pre-heated, before being introduced to the Herriott cell. To study the reaction between $CH_2OO$ and HCHO, we performed flash photolysis of flowing mixtures of $CH_2I_2$/HCHO/$O_2$/$N_2$. A small stream of $O_2$ was bubbled through the liquid $CH_2I_2$ (Sigma-Aldrich, 99% purity) before entering the reactor and the mixing ratio of $CH_2I_2$ in $O_2$ can be measured by UV absorption in a 55-cm quartz cell at 285 nm. The formaldehyde (HCHO) was prepared by heating par-aformaldehyde (Sigma-Aldrich, 95% purity) to 100–130 °C and then passed through a cold trap at −40 °C to remove coproducts such as water and other impurities, before being controlled by a mass flow controller and introduced to a 40-cm absorption cell. The mixing ratio of the gaseous HCHO in the bath gas $O_2$/$N_2$ before injection into the main reactor was determined by employing UV absorption spectra and the absorption cross section of HCHO in region 350−358 nm[43]. The Herriott mirrors and cell windows were also purged by a small stream of $N_2$. The total flow rate of over a few thousands standard cm$^3$ min$^{-1}$ was typically used at our experimental conditions. The partial pressure of each precursor in the Herriott reactor cell can be calculated according to the flow rate of each stream, mixing ratios of the pre-mixtures, and the total pressure.

**Global chemistry-transport model.** Model simulations were conducted by a 3-D global chemistry and transport model, STOCHEM-CRI[44,45] to assess the loss of HCHO by $CH_2OO$ and the subsequent formation of HCOOH throughout the troposphere. STOCHEM-CRI is a Lagrangrian model where the tropospheric layer of the atmosphere is divided into 50,000 air parcels of constant mass. The model allows chemical processes within the air parcel, alongside the emission and removal processes, to be uncoupled from the transportation of the parcel. Within the model, both transport and chemical processes are driven by meteorological archived data from the U.K. Meteorological Office (UKMO) Unified Model. The UKMO model operates across a grid resolution of 1.25° longitudes by 0.83° latitude and 12 unevenly spaced vertical levels, with the upper boundary up to 100 hPa[46]. The output data from the model run has a resolution of 5° longitude by 5° latitude and has 9 vertical levels, which span from the surface up to an altitude of 16 km. The chemical mechanism used in STOCHEM is the common representative intermediate mechanism version 2 and reduction 5 (CRI v2-R5). The details of the CRI v2-R5 mechanism can be found in Jenkin et al.[47], Watson et al.[48], Utembe et al.[49] and Jenkin et al.[50] Further amendments to the chemical mechanism with the addition of Criegee field are reported by Chhantyal-Pun et al.[37] Emissions of the species within STOCHEM are categorized into surface emissions, stratospheric sources, and the three-dimensional emissions. Surface emissions from ocean, soil, vegetation, and biomass burning are distributed using monthly two-dimensional source maps with a resolution of 5° longitude by 5° latitude[51]. The total emissions for nitrogen oxides, carbon monoxide, and nonmethane volatile organic compounds (NMVOCs) included in the model were adapted from the Precursors of Ozone and their Effects in the Troposphere (POET) inventory[52]. Further details of the STOCHEM emissions inventory for all species can be found in Khan et al.[53] Wet deposition and dry deposition are the two main techniques used in STOCHEM to remove chemical species from air parcels at the boundary layer. Dry deposition is accounted for using a resistance approach in STOCHEM and the rate at which it occurs depends on whether a Lagrangian air parcel is treated as being above land or the ocean. The dry deposition velocities used in STOCHEM were adapted from the annual mean values calculated from the MATCH global model[54]. Wet deposition is the elimination of soluble species through dissolution in precipitation. These dissolved substances may have environmental origins or may mix with precipitation as it falls. To determine wet deposition loss rates, wet deposition equations in STOCHEM combine species-dependent scavenging coefficients taken from Penner et al.[55] with scavenging profiles and precipitation rates.

The base case simulation conducted was based on the reference condition described in Chaantyal-Pun et al.[37] with additional amendment of the CRI mechanism by adding HCOOH + $CH_2OO$ reaction ($k = 0.7 \times 10^{-10}$ cm$^3$ molecule$^{-1}$ s$^{-1}$)[56]. Another simulation was conducted with inclusion of HCHO + $CH_2OO$ reaction yielding of HCOOH + HCHO (43%) and CO + $H_2O$ + HCHO (57%) in STOCHEM-CRI. Both simulations were conducted

with 1998 meteorology data for a period of 24 months with the first 12 allowing the model to spin up. Analysis were performed on the subsequent 12 months of data.

## Data availability

The data supporting the findings of this study are available within the article and its Supplementary Information and from the corresponding author upon reasonable request.

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

## Acknowledgements

We thank Dr. Kaito Takahashi for reading the manuscript and helpful discussion. This project is supported by National Science and Technology Council, Taiwan (grant No. 111-2112-M-001-067 and grant No. 111-2639-M-A49-001-ASP) and Academia Sinica.

## Author contributions

P.-L.L. built the experimental system, performed the experiments, and analyzed the data. I-Y.C. performed concentration measurements of formaldehyde and analyzed the data. M.A.H.K. and D.E.S. performed the simulations with the global chemistry-transport model. P.-L.L. supervised the project and wrote the paper.

## Competing interests

The authors declare no competing interests.
