## [Peer Review File · Communications Chemistry]

Reviewers' comments:

Reviewer #1 (Remarks to the Author):

The authors present the results of a gas phase study of the kinetics and product yields of the reaction between the Criegee intermediate CH₂OO (formaldehyde oxide) and HCHO (formaldehyde) under pseudo-first-order conditions using laser flash photolysis of CH₂I₂/O₂/N₂/HCHO mixtures with detection of CH₂OO and HCOOH (formic acid) and CO (carbon monoxide) products by mid-infrared dual comb spectroscopy. Experiments were performed over a range of conditions relevant to the atmosphere, and temperature-dependent results for the kinetics and product yields are reported, with some discussion of the potential atmospheric significance of the results.

The experimental setup is state-of-the-art and the results are well described and presented in the manuscript. There has been significant interest in Criegee intermediate chemistry in recent years, and an experimental study of the temperature-dependent kinetics and product yields of CH₂OO + HCHO will be of interest to the atmospheric chemistry community.

The authors report significant product of formic acid from the title reaction. There has been much uncertainty in the literature over measurements of formic acid in the troposphere. Atmospheric models typically underestimate atmospheric observations of formic acid, with consequences for accurate modelling of air quality and climate. The authors report that the results presented in this work may explain observations of HCOOH in the upper tropospheric/lower stratosphere (UT/LS) and provide some relatively simple calculations to support this. However, isoprene has a lifetime of ~2 hours in the troposphere owing to rapid reaction with OH and it is not clear that there would be sufficient transport of isoprene to the UT/LS to support production of CH₂OO from isoprene, and subsequently of HCOOH from the title reaction. It may be the case that other unsaturated alkenes are present in the UT/LS at sufficient concentration to generate CH₂OO, and thus HCOOH, but this is not discussed. The atmospheric implications presented are somewhat speculative and would be more convincing through use of a global atmospheric chemistry model and detailed comparisons to observed HCOOH concentrations. For a high impact paper based on the potential for HCOOH production this should be expected.

Minor comments are listed below:

The authors use 'rate constant' and 'rate coefficient' throughout. Consistency is required, and use of 'rate coefficient' is preferred.

Abstract: It would be helpful to include details of results for $k(296\text{ K})$, Arrhenius parameters, and branching ratios.

Lines 35-38: Some elaboration and further description of the role of HCOOH in the atmosphere, and the impacts of model underestimations of HCOOH is required. Model underestimations should also be quantified.

Line 42: What is the typical lifetime of HCOOH in the atmosphere?

Line 44: How much isoprene is typically oxidised by OH vs ozone?

Line 48: 'can be ...' to 'which can be ...'.

Lines 52 and 56: What are the uncertainties on the yields given?

Line 58: 'OH scavenger' to 'an OH scavenger' or 'OH scavengers'.

Line 60: 'low concentration of water and OH radical' to 'low concentrations of water and OH radicals'.

Lines 61-65: Please provide a brief summary of the evidence provided by Long et al. provide for the impact of CH₂OO + HCHO in the UT/LS.

Line 66: '... is shown in...' to '... are shown in ...'.

Line 108: 'with kinetic model' to 'with a kinetic model'.

Line 121: 'with single exponential function' to 'with a single exponential function'.

Lines 138-144: What is the result for the pre-exponential factor, A? This is essential to implementation of results from this work in atmospheric models. Is the value for the rate coefficient at 296 K the average of experimental results at this temperature, or the result at 296 K from the parameterisation using the Arrhenius equation? How does the value for the activation energy compare to the theoretical value? Some further discussion of the differences between experimental results and theory is required.

Line 168: Typo in 'employed'.

Lines 186-188: How significant was the production of CO from HCHO photolysis? How do uncertainties in quantum yields for HCHO photolysis impact the results reported here?

Line 198: It would be helpful to show the time-dependence of the HCOOH and CO signals in the main text. Supplementary Figure 7 shows significant slow production of HCOOH on the millisecond timescale, which appears to be well characterised by the model. What is the source? The only reactions in the model producing HCOOH are CH₂OO + HCHO and relaxation of excited HCOOH, is there significant relaxation of excited HCOOH at long times? Do the kinetics of the slow growth support production from excited states? Is there any theory to suggest significant production of excited HCOOH from CH₂OO + HCHO? The kinetics of CH₂OO suggest there cannot be direct production from CH₂OO on the millisecond timescale. Is there potential for production of HCOOH via other secondary chemistry/processes that are not described in the model?

Line 220: Is the temperature-dependence of CH₂OO + HCOOH well characterised in the literature? How do uncertainties in the kinetics of this reaction impact the results of this work?

Line 227: Is the pressure broadening well-characterised? What are the uncertainties in these parameters, and how do they impact the results?

Line 260: Solar photolysis of HCHO should also be considered.

Line 265: Mixing ratios of 1 ppm HCHO in the atmosphere are somewhat high. Calculations using a range of lower values, or ideally using an atmospheric model as suggested above, would be helpful.

Line 270: Is the value of 123 s⁻¹ for reaction with the water monomer (H₂O) or dimer ((H₂O)₂)?

Line 284: The statement 'isoprene might more effectively ...' could be established readily using an atmospheric model to avoid conjecture.

Line 306: What values are used for OH concentrations?

Figure 8: Such plots are often given with altitude on the y-axis.

Line 349: 'in the absence of the OH scavenger' to 'in the absence of OH scavengers'.

Reviewer #2 (Remarks to the Author):

Manuscript ID COMMSCHEM-23-0104-T, entitled " Direct Gas-Phase Formation of HCOOH through Reaction of Criegee Intermediates with Formaldehyde ". The work is very important and interesting for

understanding the formation of formic acid via CH₂OO. The present work provides a new source for formic acid in the atmosphere; this helps to resolve the difference in the concentration of formic acid between field measurements and atmospheric modeling. However, there are some key issues that should be addressed before publication.

(1) The key issue is the big difference between theoretical results and experimental data. The experimental result is at least 10 times than that of the theoretical results. The big difference shows that theoretical results are unreliable, or experimental data is unreliable. It is particularly noted that the reaction is simple from the mechanistic point of view. In addition, the rate determining step is the tight transition state from theoretical point of view at 298 K. Therefore, the enthalpy of activation at 0 K is the most parameter because recrossing and tunneling effects are negligible in reference 14. If the experimental data is right, the error bar of enthalpy of activation at 0 K for theoretical method is about 1.5 kcal/mol. However, there are lots of results that have shown that the method in reference 14 can obtain quantitative rate constants; this requires the error bar below 0.2 kcal/mol. Please read the references (J. Am. Chem. Soc. 2016, 138, 14409-14422.; Proc. Natl. Acad. Sci. USA 2018, 115, 6135-6140.; Phys. Chem. Chem. Phys. 2022, 24, 24759-24766.; Phys. Chem. Chem. Phys., 2022, 24, 13066-13073.).

(2) How to eliminate the effects of the CH₂OO + HCOOH reaction on the CH₂OO + HCHO reaction, when the rate constant of CH₂OO + HCHO is measured because the CH₂OO + HCOOH reaction is very fast.

(3) In the Discussion Section, there are key references missed such as (J. Am. Chem. Soc. 2016, 138, 14409-14422.; J. Am. Chem. Soc. 2021, 143, 8402-8413.) because the rate ratio has been done in the references. In addition, HO₂ can make certain contribution to the sink of HCHO, which should be discussed (J. Am. Chem. Soc. 2022.144, 19910-19920.).

(4) There are some minor issues. For example, Lines 303-304, 2 is subscript.

Response and revisions to referees' comments

We appreciate very much the valuable comments by the reviewers. We have revised the manuscript accordingly and hope that this manuscript is now acceptable for publication in Communications Chemistry. For convenience, we listed the response and revisions in blue font color after each comment.

Reviewer #1 (Remarks to the Author):

The authors present the results of a gas phase study of the kinetics and product yields of the reaction between the Criegee intermediate CH_2OO (formaldehyde oxide) and HCHO (formaldehyde) under pseudo-first-order conditions using laser flash photolysis of $\text{CH}_2\text{I}_2/\text{O}_2/\text{N}_2/\text{HCHO}$ mixtures with detection of CH_2OO and HCOOH (formic acid) and CO (carbon monoxide) products by mid-infrared dual comb spectroscopy. Experiments were performed over a range of conditions relevant to the atmosphere, and temperature-dependent results for the kinetics and product yields are reported, with some discussion of the potential atmospheric significance of the results.

The experimental setup is state-of-the-art and the results are well described and presented in the manuscript. There has been significant interest in Criegee intermediate chemistry in recent years, and an experimental study of the temperature-dependent kinetics and product yields of $\text{CH}_2\text{OO} + \text{HCHO}$ will be of interest to the atmospheric chemistry community.

The authors report significant product of formic acid from the title reaction. There has been much uncertainty in the literature over measurements of formic acid in the troposphere. Atmospheric models typically underestimate atmospheric observations of formic acid, with consequences for accurate modelling of air quality and climate. The authors report that the results presented in this work may explain observations of HCOOH in the upper tropospheric/lower stratosphere (UT/LS) and provide some relatively simple calculations to support this. However, isoprene has a lifetime of ~ 2 hours in the troposphere owing to rapid reaction with OH and it is not clear that there would be sufficient transport of isoprene to the UT/LS to support production of CH_2OO from isoprene, and subsequently of HCOOH from the title reaction. It may be the case that other unsaturated alkenes are present in the UT/LS at sufficient concentration to generate CH_2OO , and thus HCOOH , but this is not discussed. The atmospheric implications presented are somewhat speculative and would be more convincing through use of a global atmospheric chemistry model and detailed comparisons to observed HCOOH concentrations. For a high impact paper based on the potential for HCOOH production this should be expected.

<Response & Revision 1-1>

We thank the reviewer for these suggestions. We have employed a global chemistry-transport

model, STOCHEM-CRI, to evaluate the atmospheric implications of the reaction $\text{CH}_2\text{OO} + \text{HCHO}$.

We agree with the reviewer that the lifetime of isoprene is short, which inhibits sufficient transport of isoprene to the UT/LS to support the production of CH_2OO from isoprene. In our recent publication (Chhantyal-Pun et al., ACS Earth Space Chem. 2019, 3, 2363-2371), we showed that a significant amount of CH_2OO (~ 400 molecules/ cm^3) can be formed in the upper troposphere. In that publication, we also showed that CH_2OO was the main contributor to all Criegee intermediates in the upper troposphere because of lower H_2O concentrations. CH_2OO can also be formed from the ozonolysis of the fragments of isoprene ozonolysis, resulting in a long-range transport, so we found a large amount of CH_2OO in the upper troposphere. We used the same Criegee field in the model calculation for this study. We have also included the loss of CH_2OO by HCOOH in the current study. According to the results of the global chemistry-transport modelling, the percent loss of CH_2OO by HCHO is obtained by up to 6% and it can cause over 2% percent increasing of HCOOH during December-January-February months in the upper troposphere.

The main limitation of the model study is the uncertainty of the alkene emission inventory which has been discussed in the revised version nicely. The updating of the emission inventory is important which could make the reaction $\text{CH}_2\text{OO} + \text{HCHO}$ even more important contributor for the HCOOH formation, particularly in upper troposphere.

Minor comments are listed below:

The authors use 'rate constant' and 'rate coefficient' throughout. Consistency is required, and use of 'rate coefficient' is preferred.

<Response & Revision 1-2>

We thank the reviewer for pointing this out to us. We have revised it accordingly.

Abstract: It would be helpful to include details of results for $k(296 \text{ K})$, Arrhenius parameters, and branching ratios.

<Response & Revision 1-3>

We thank the reviewer for this suggestion. We have added the results for $k(296 \text{ K})$, Arrhenius parameters, and branching ratios in the abstract.

Lines 35-38: Some elaboration and further description of the role of HCOOH in the atmosphere, and the impacts of model underestimations of HCOOH is required. Model underestimations should also be quantified.

Line 42: What is the typical lifetime of HCOOH in the atmosphere?

<Response & Revision 1-4>

We have revised the sentence. “In the atmosphere, HCOOH can be produced from direct biogenic emissions, biomass burning as well as gas-phase and multi-phase chemical reactions; and it can be removed mainly through wet and dry deposition¹⁻⁶. The mixing ratio of atmospheric HCOOH is typically observed from sub-100 pptv to a few ppbv levels and the lifetimes of HCOOH are estimated to be 1–2 days and 1–2 weeks in the boundary layer and upper troposphere, respectively.^{3,5} Thanks to satellite observation techniques^{1,5,6}, the atmospheric abundance of HCOOH even in remote areas can be in-situ monitored. However, the current chemistry-climate models still cannot fully expound the unexpected high levels of observed HCOOH in the atmosphere. Typically, the HCOOH concentrations from modeling results are a factor of 2–5 times lower than that from observations^{1,3,4,6}.”

Line 44: How much isoprene is typically oxidised by OH vs ozone?

According to the chamber experiments of isoprene oxidation⁹, ~87 % and 10 % of the isoprene can be oxidised by OH and ozone, respectively. And nearly 30 % and up to 40 % of the global annual production of atmospheric HCOOH from gas-phase reactions can be contributed from the OH-initiated oxidation of isoprene and isoprene ozonolysis, respectively.

Line 48: ‘can be ...’ to ‘which can be ...’.

<Response & Revision 1-5>

We have revised it accordingly.

Lines 52 and 56: What are the uncertainties on the yields given?

<Response & Revision 1-6>

We have added the uncertainties on the yields.

Line 58: ‘OH scavenger’ to ‘an OH scavenger’ or ‘OH scavengers’.

Line 60: ‘low concentration of water and OH radical’ to ‘low concentrations of water and OH radicals’.

<Response & Revision 1-7>

We have revised it accordingly.

Lines 61-65: Please provide a brief summary of the evidence provided by Long et al. provide for the impact of CH₂OO + HCHO in the UT/LS.

<Response & Revision 1-8>

We have rewritten this sentence and added a comparison of the different theoretical investigations.

Line 66: ‘... is shown in...’ to ‘... are shown in ...’.

Line 108: ‘with kinetic model’ to ‘with a kinetic model’.

Line 121: ‘with single exponential function’ to ‘with a single exponential function’.

<Response & Revision 1-9>

We have revised it accordingly.

Lines 138-144: What is the result for the pre-exponential factor, A? This is essential to implementation of results from this work in atmospheric models. Is the value for the rate coefficient at 296 K the average of experimental results at this temperature, or the result at 296 K from the parameterisation using the Arrhenius equation? How does the value for the activation energy compare to the theoretical value? Some further discussion of the differences between experimental results and theory is required.

<Response & Revision 1-10>

We have rewritten this part and added a comparison of experimental and theoretical results of the rate coefficient for the reactions of CH₂OO with HCHO, CH₃CHO, and CH₃COCH₃, as shown in Supplementary Table 2. By analyzing the data from 7 experimental sets at 296 K, $k_{\text{CH}_2\text{OO}+\text{HCHO}}$ was obtained to be $(4.11 \pm 0.25) \times 10^{-12} \text{ cm}^3 \text{ molecule}^{-1} \text{ s}^{-1}$ at 296 K which is 15 times smaller than the theoretical value reported by Long *et al.*¹⁹, but it is ~5 times larger than the predicted value from Jalan *et al.*¹⁷ The temperature dependence of the rate coefficient was fitted to an Arrhenius expression with a pre-exponential constant of $(1.91 \pm 0.15) \times 10^{-13} \text{ cm}^3 \text{ molecule}^{-1} \text{ s}^{-1}$ and an activation energy of $(-1.81 \pm 0.04) \text{ kcal mol}^{-1}$. Although this activation energy for CH₂OO + HCHO is not in agreement with the theoretical results, it is comparable with $(-2.2 \pm 0.7) \text{ kcal mol}^{-1}$ measured by Elsamra *et al.*²³ for reactions involving other ketones: CH₂OO + CH₃CHO and CH₂OO + CH₃COCH₃.

Line 168: Typo in ‘employed’.

<Response & Revision 1-11>

We have revised it accordingly.

Lines 186-188: How significant was the production of CO from HCHO photolysis? How do uncertainties in quantum yields for HCHO photolysis impact the results reported here?

<Response & Revision 1-12>

The fractional yield of photodissociation of HCHO was obtained to be ~0.02% at 248 nm in the experiments and approximately 8×10^{11} molecule cm^{-3} of CO could be produced when the $[\text{HCHO}]_0$ of 4×10^{15} molecule cm^{-3} was used.

Line 198: It would be helpful to show the time-dependence of the HCOOH and CO signals in the main text. Supplementary Figure 7 shows significant slow production of HCOOH on the millisecond timescale, which appears to be well characterised by the model. What is the source? The only reactions in the model producing HCOOH are $\text{CH}_2\text{OO} + \text{HCHO}$ and relaxation of excited HCOOH, is there significant relaxation of excited HCOOH at long times? Do the kinetics of the slow growth support production from excited states? Is there any theory to suggest significant production of excited HCOOH from $\text{CH}_2\text{OO} + \text{HCHO}$? The kinetics of CH_2OO suggest there cannot be direct production from CH_2OO on the millisecond timescale. Is there potential for production of HCOOH via other secondary chemistry/processes that are not described in the model?

<Response & Revision 1-13>

We thank the reviewer for these suggestion. We have moved the figures of the time-dependence of the HCOOH and CO signals into the main text. According to the enthalpy profiles of the reaction $\text{CH}_2\text{OO} + \text{HCHO}$ based on the theoretical investigations, the $\text{CH}_2\text{OO} + \text{HCHO}$ might generate the the higher-energy *cis* conformer of formic acid. Therefore, $\text{HCOOH}^\#$ might include the *trans*-HCOOH ($v > 0$) and the higher-energy *cis* conformer of formic acid. The relaxations of the energized $\text{HCOOH}^\#$, including *trans*-HCOOH ($v > 0$) to *trans*-HCOOH ($v = 0$) and the *cis*-HCOOH to *trans*-HCOOH conversions³⁰ were also taken into account in the model.

Line 220: Is the temperature-dependence of $\text{CH}_2\text{OO} + \text{HCOOH}$ well characterised in the literature? How do uncertainties in the kinetics of this reaction impact the results of this work?

<Response & Revision 1-14>

The temperature-dependent rate coefficients of the reaction $\text{CH}_2\text{OO} + \text{HCOOH}$ have been determined by Peltola *et al.*³¹ and the rate coefficient is $(1.0 \pm 0.03) \times 10^{-10}$ cm^3 molecule⁻¹ s⁻¹ at 296 K. We assumed that the rate coefficients of the reactions $\text{CH}_2\text{OO} + \text{HCOOH}$ and $\text{CH}_2\text{OO} + \text{HCOOH}^\#$ are the same. By adding these reactions in the model, we observed that the y_{HCOOH} increased by ~10% for experimental conditions with $[\text{CH}_2\text{OO}]_0 = (3.2\text{--}3.8) \times 10^{13}$ molecules cm^{-3} , $[\text{HCHO}]_0 = (2.2\text{--}4.0) \times 10^{15}$ molecule cm^{-3} , and $T = 283\text{--}313$ K. The uncertainty for the rate coefficients of the reaction $\text{CH}_2\text{OO} + \text{HCOOH}$ is small and it is neglectable for determining the y_{HCOOH} .

Line 227: Is the pressure broadening well-characterised? What are the uncertainties in these parameters, and how do they impact the results?

<Response & Revision 1-15>

To estimate the concentration of the HCOOH in the reaction system, several transition lines of the ν_6 band of the *trans*-HCOOH near 1128 cm^{-1} were measured with dual-comb spectroscopy and analyzed with well-characterized spectral parameters²⁷⁻²⁹. Fig. R1 shows the comparison of the measured and simulated spectra of HCOOH. The standard deviation of residuals between observed and simulated spectra is typically 5~7%.

Fig. R1 The absorbance spectra of the gaseous mixtures of (a) HCOOH/O₂ (0.38/40.12 Torr, $P_T = 40.5$ Torr, 296 K) and (b) HCOOH/O₂ (0.38/79.82 Torr, $P_T = 80.2$ Torr, 296 K). Here, a 15-cm sample cell was used.

Line 260: Solar photolysis of HCHO should also be considered.

<Response & Revision 1-16>

We have added it in the text.

Line 265: Mixing ratios of 1 ppm HCHO in the atmosphere are somewhat high. Calculations using a range of lower values, or ideally using an atmospheric model as suggested above, would be helpful.

<Response & Revision 1-17>

In the ambient atmosphere, the concentrations of gaseous HCHO can be observed up to sub-ppm and a few ppm levels, respectively, in outdoor and indoor environments^{35,36}. Mixing ratios of 1 ppm HCHO in the atmosphere might be obtained in the indoor environments. We have also employed a global chemistry-transport model to evaluate the atmospheric implications of the reaction $\text{CH}_2\text{OO} + \text{HCHO}$.

Line 270: Is the value of 123 s^{-1} for reaction with the water monomer (H_2O) or dimer ($(\text{H}_2\text{O})_2$)?

<Response & Revision 1-18>

The effective first-order rate coefficient is estimated to be $\sim 143 \text{ s}^{-1}$ which is comparable to that ($\sim 123 \text{ s}^{-1}$) of CH_2OO with water monomer at RH of 50 % and 296 K.

Line 284: The statement ‘isoprene might more effectively ... ’ could be established readily using an atmospheric model to avoid conjecture.

Line 306: What values are used for OH concentrations?

Figure 8: Such plots are often given with altitude on the y-axis.

Line 349: ‘in the absence of the OH scavenger’ to ‘in the absence of OH scavengers’.

<Response & Revision 1-19>

We have rewritten this part and added the simulations with a global chemistry-transport model to evaluate the atmospheric implications of the reaction $\text{CH}_2\text{OO} + \text{HCHO}$.

Reviewer #2 (Remarks to the Author):

Manuscript ID COMMSCHEM-23-0104-T, entitled " Direct Gas-Phase Formation of HCOOH through Reaction of Criegee Intermediates with Formaldehyde ". The work is very important and interesting for understanding the formation of formic acid via CH_2OO . The present work provides a new source for formic acid in the atmosphere; this helps to resolve the difference in the concentration of formic acid between field measurements and atmospheric modeling. However, there are some key issues that should be addressed before publication.

(1) The key issue is the big difference between theoretical results and experimental data. The experimental result is at least 10 times than that of the theoretical results. The big difference shows that theoretical results are unreliable, or experimental data is unreliable. It is particularly noted that the reaction is simple from the mechanistic point of view. In addition, the rate determining step is the tight transition state from theoretical point of view at 298 K. Therefore, the enthalpy of activation at 0 K is the most parameter because recrossing and tunneling effects are negligible in reference 14. If the experimental data is right, the error bar of enthalpy of activation at 0 K for theoretical method is about 1.5 kcal/mol. However, there are lots of results

that have shown that the method in reference 14 can obtain quantitative rate constants; this requires the error bar below 0.2 kcal/mol. Please read the references (J. Am. Chem. Soc. 2016, 138, 14409-14422.; Proc. Natl. Acad. Sci. USA 2018, 115, 6135-6140.; Phys. Chem. Chem. Phys. 2022, 24, 24759-24766.; Phys. Chem. Chem. Phys., 2022, 24, 13066-13073.).

<Response & Revision 2-1>

We thank the reviewer for pointing out this issue concerning theoretical results. We found two recent theoretical publications; one is from Jalan *et al.* (*Phys. Chem. Chem. Phys.* **15**, 16841–16852 (2013)) and another is the one suggested by the reviewers: Long *et al.* (*J. Am. Chem. Soc.* **143**, 8402–8413 (2021)). When we evaluated the theoretical $k_{\text{CH}_2\text{OO}+\text{HCHO}}$ from these two publications, we found a large difference between them. The 298 K rate coefficient reported by Long *et al.*, $6.2 \times 10^{-11} \text{ cm}^3 \text{ molecule}^{-1} \text{ s}^{-1}$, was ~ 75 times larger than the value predicted by Jalan *et al.*, $8.3 \times 10^{-13} \text{ cm}^3 \text{ molecule}^{-1} \text{ s}^{-1}$. Arrhenius plot of the experimental and theoretical rate coefficients are given in Figure R2. Our experimental values lie in between the two theoretical predictions. Because we are not experts in theoretical calculations, we cannot judge the results from the two different theoretical calculations. We have confirmed our experimental conditions to the best we can and provided the error bar of our estimates in the figure. We believe future theoretical studies can remedy this difference between experimental and theoretical results.

Following the suggestion by the referee, we have added a description concerning various previous theoretical investigations on the main text and Supplementary Note 1. We also added a comparison of experimental and theoretical results of the rate coefficient for the reactions of CH₂OO with HCHO, CH₃CHO, and CH₃COCH₃, as shown in Supplementary Table 2. By analyzing the data from 7 experimental sets at 296 K, $k_{\text{CH}_2\text{OO}+\text{HCHO}}$ was obtained to be $(4.11 \pm 0.25) \times 10^{-12} \text{ cm}^3 \text{ molecule}^{-1} \text{ s}^{-1}$ at 296 K which is 15 times smaller than the theoretical value reported by Long *et al.*¹⁹, but it is ~ 5 times larger than the predicted value from Jalan *et al.*¹⁷ The temperature dependence of the rate coefficient was obtained and described by an Arrhenius expression with a pre-exponential constant of $(1.91 \pm 0.15) \times 10^{-13} \text{ cm}^3 \text{ molecule}^{-1} \text{ s}^{-1}$ and an activation energy of $(-1.81 \pm 0.04) \text{ kcal mol}^{-1}$. Although this experimental activation energy for CH₂OO + HCHO is not in agreement with the theoretical results, it is comparable with $(-2.2 \pm 0.7) \text{ kcal mol}^{-1}$ measured by Elsamra *et al.*²³ for reactions involving other ketones: CH₂OO + CH₃CHO and CH₂OO + CH₃COCH₃.

Fig. R2 Arrhenius plots of the rate coefficients for the reaction $\text{CH}_2\text{OO} + \text{HCHO}$. Each red square represents experimental data obtained in this work. The blue line represents the value predicted by Jalan *et al.* and the black line represents the value predicted by Long *et al.*

(2) How to eliminate the effects of the $\text{CH}_2\text{OO} + \text{HCOOH}$ reaction on the $\text{CH}_2\text{OO} + \text{HCHO}$ reaction, when the rate constant of $\text{CH}_2\text{OO} + \text{HCHO}$ is measured because the $\text{CH}_2\text{OO} + \text{HCOOH}$ reaction is very fast.

<Response & Revision 2-2>

In the experiments for determining the rate coefficients, we used a lower concentration of CH_2OO , $[\text{CH}_2\text{OO}]_0 = (5.5\text{--}8.6) \times 10^{12}$ molecules cm^{-3} , and set $[\text{HCHO}]_0 \gg \gg [\text{CH}_2\text{OO}]_0$, the effects of the $\text{CH}_2\text{OO} + \text{HCOOH}$ reaction on the rate coefficients of $\text{CH}_2\text{OO} + \text{HCHO}$ reaction is hence small and might be neglectable for determining the rate coefficients. Please see the Supplementary Figure 4. We have employed the kinetic model with and without the addition of the reaction $\text{CH}_2\text{OO} + \text{HCOOH}$ to analyze the time traces of CH_2OO . The second-order rate coefficient $k_{\text{CH}_2\text{OO}+\text{HCHO}}$, corresponding to the fitted slopes derived by using single-exponential and model fits, were consistent with each other, supporting the feasibility of kinetic analysis under pseudo-first order conditions.

(3) In the Discussion Section, there are key references missed such as (J. Am. Chem. Soc. 2016, 138, 14409-14422.; J. Am. Chem. Soc. 2021, 143, 8402-8413.) because the rate ratio has been done in the references. In addition, HO_2 can make certain contribution to the sink of HCHO ,

which should be discussed (J. Am. Chem. Soc. 2022.144, 19910-19920.).

(4) There are some minor issues. For example, Lines 303-304, 2 is subscript.

<Response & Revision 2-3>

We have rewritten this part and added the simulations with a global chemistry-transport model to evaluate the atmospheric implications of the reaction $\text{CH}_2\text{OO} + \text{HCHO}$. Please see

<Response & Revision 1-1>.

REVIEWERS' COMMENTS:

Reviewer #1 (Remarks to the Author):

The authors have addressed the concerns raised in the initial review and, in my opinion, have significantly improved the manuscript through the use of a global model to assess the impacts of the title reaction.

Reviewer #2 (Remarks to the Author):

I think that the authors have well addressed my concern.

Response and revisions to referees' comments

Reviewer #1 (Remarks to the Author):

The authors have addressed the concerns raised in the initial review and, in my opinion, have significantly improved the manuscript through the use of a global model to assess the impacts of the title reaction.

Reviewer #2 (Remarks to the Author):

I think that the authors have well addressed my concern.

We appreciate very much the valuable comments by the reviewers.